# Structural Analysis and Substrate Specificity of D-Carbamoylase from *Pseudomonas*

**DOI:** 10.3390/biotech13040040

**Published:** 2024-10-03

**Authors:** Marina Paronyan, Haykanush Koloyan, Hovsep Aganyants, Artur Hambardzumyan, Tigran Soghomonyan, Sona Avetisyan, Sergey Kocharov, Henry Panosyan, Vehary Sakanyan, Anichka Hovsepyan

**Affiliations:** 1Scientific and Production Center ”Armbiotechnology”, National Academy of Sciences of Armenia, Yerevan 0056, Armenia; paronyan_marina@mail.ru (M.P.); sonulik_79@mail.ru (S.A.); anichka_h@mail.ru (A.H.); 2Scientific Technological Centre of Organic and Pharmaceutical Chemistry SNPO, National Academy of Sciences of Armenia, Yerevan 0014, Armenia; 3Faculty of Science and Technique, Nantes University, 44035 Nantes, France

**Keywords:** D-carbamoylase, homology modeling, molecular docking, molecular dynamics, N-carbamoyl-D-alanine, N-carbamoyl-D-tryptophan, inclusion bodies, *Pseudomonas*

## Abstract

The synthesis of enantiomeric forms of D-amino acids can be achieved by a two-step “hydantoinase process” based on the sequential catalysis of substrates by specific enzymes, D-carbamoylase and D-hydantoinase. Here, we describe the structural features of D-carbamoylase from *Pseudomonas*, the encoded gene of which was chemically synthesized and cloned into *Escherichia coli*. A significant fraction of the overexpressed recombinant protein forms insoluble inclusion bodies, which are partially converted to a soluble state upon treatment with N-lauroylsarcosine or upon incubation of cells at 28 °C. Purified His-tagged protein exhibits the highest activity towards N-carbamoyl-D-alanine and N-carbamoyl-D-tryptophan. Comprehensive virtual analysis of the interactions of bulky carbamylated amino acids with D-carbamoylase provided valuable information. Molecular docking analysis revealed the location of the substrate binding site in the three-dimensional structure of D-carbamoylase. Molecular dynamics simulations showed that the binding pocket of the enzyme in complex with N-carbamoyl-D-tryptophan was stabilized within 100 nanoseconds. The free energy data showed that Arg176 and Asn173 formed hydrogen bonds between the enzyme and substrates. The studies of D-carbamoylases and the properties of our previously obtained D-hydantoinase suggest the possibility of developing a harmonized biotechnological process for the production of new drugs and peptide hormones.

## 1. Introduction

The production of optically pure D-amino acids and their derivatives by the enantioselective ring opening of hydantoins and the subsequent addition to a decarbamoylated substrate is known as the “hydantoinase process” (Figure 1) [1,2]. This advantageous biotechnological process is based on the use of the enzymes D-hydantoinase and D-carbamoylase, where D-cabamoylase is the rate-limiting enzyme in a two-step reaction system [3,4]. D-hydantoinase (EC 3.5.2.2) catalyzes the stereospecific cleavage of chemically synthesized 5-substituted hydantoin to form the substrate for D-carbamoyl amino acid. Hydantoin (also called imidazolidine-2,4-dione) [5] and dihydropyrimidine are alternative substrates for hydantoinase. D-carbamoylase (N-carbamoyl-D-amino acid amidohydrolase) (EC 3.5.1.77) hydrolyzes the resulting D-carbamoyl amino acid to form the desired D-amino acid.

The first mentions of eukaryotic organisms exhibiting the ability to break down hydantoin date back to 1926 [6], and this information was then used to study the enzymes involved in the catalytic process [7,8]. The first mention of bacterial D-carbamoylase from *Agrobacterium* was noted in the synthesis of D-amino acids in 1979 [9]. Then, D-carbamoylases were described in other microorganisms, including *Arthrobacter* sp. [10], *Pseudomonas* sp. [11], *Blastobacter* sp. [12], and *Comamonas* sp. [13]. Subsequent studies allowed the cloning and expression of the foreign bacterial D-carbamoylase gene and the characterization of the overexpressed protein in *E. coli* cells [10,12,14,15]. The possibility of enantioselective racemization of hydantoin derivatives using two enzymes naturally aroused applied interest in the biotechnological production of optically pure D-amino acids [16].

Biochemical studies have shown that D-carbamoylase can have different structures: homodimers or homotetramers with subunits of 32–40 kDa [4] and homotrimers with subunits of 40 kDa [13]. The optimal pH for D-carbamoylase enzymatic activity ranges from 7.0 to 9.0 [13,15]. Enzyme activity is inhibited by a number of metal ions; however, the enzyme is reactivated by reducing agents, and chelating agents do not affect its activity, suggesting that cations do not play a significant role in enzyme activity [17,18]. Using random and site-directed mutagenesis of D-carbamoylase from the mesophilic bacterium *Agrobacterium* sp. KNK712, thermostable mutants up to 71.4 °C were obtained [19,20]. These data indicated the possibility of modulating the catalytic functions of bacterial D-carbamoylase to adapt it to the technological process.

Crystal structures of wild-type and mutant D-carbamoylases were generated for the protein from *Nitratireductor indicus* [21], *Agrobacterium* sp. KNK712 [22], and *A. radiobacter* [23], which are available in the Protein Data Bank. The conserved amino acid sequence of Cys-Glu-Lys indicates its critical role in the catalytic activity of the protein in bacteria. It is assumed that the reaction mechanism of the Cys-Glu-Lys triad occurs through the process of acylation–deacylation [22,23].

The search for carbamolyase activity in thermostable bacteria allowed us to detect L-carbamoylase in a thermostable bacterium *Bacillus stearothermophilus*, but the enzyme did not meet the requirements for the production of D-amino acids [24]. However, we later also cloned and studied in detail the gene encoding D-hydantoinase from the thermophilic bacterium *Geobacillus stearothermophilus* as a candidate enzyme for the first step of the “hydantoinase process” [25,26].

In this study, we show that the D-carbamoylase gene of the *Pseudomonas* sp. strain KNK003A synthesizes an enzyme with activity towards bulky N-carbamoylated amino acids in *E. coli* host cells. Structural analysis of the enzyme’s three-dimensional structure provides important information that is useful for the production of the desired D-amino acid.

## 2. Materials and Methods

### 2.1. Reagents and Substrates

The amino acid substrates N-carbamoyl-D-alanine, N-carbamoyl-D-leucine, N-carbamoyl-D-valine, N-carbamoyl-D-phenylalanine were synthesized in the Scientific Technological Centre of Organic and Pharmaceutical Chemistry. The description of the N-carbamoyl-D-amino acids we synthesized is presented in Appendix A. Melting points were determined using a Boetius PHMK 76/0904 hot-stage microscope (Veb Analytik, Dresden, Germany). The IR spectra were recorded using a Nicolet (Thermo, Waltham, MA, USA) FT-IR spectrometer (Scotia, New York, NY, USA). 1H and 13C NMR spectra were recorded on a Varian Mercury-300 VX spectrometer (Mundelein, IL, USA) in DMSO-d6/CCl4, 1:3. Chemical shifts are reported in δ values (ppm) relative to tetramethylsilane as an internal standard. Coupling constants (J values) are given in Hertz (Hz). The signals are reported as follows: s (singlet), d (doublet), dd (double doublet), spt (septet), m (multiplet), and br (broad). All the chemicals used were of analytical or reagent grade. Carbamoyl derivatives were prepared using the D-amino acids from Reanal (Budapest, Hungary). To confirm the structure of the synthesized molecules, 1H and 13C NMR analyses were carried out (Appendix A). Other N-carbamoyl derivatives of amino acids were purchased from Sigma-Aldrich (St. Louis, MO, USA). All the chemicals were of analytical grade.

The Gibson Assembly^®^ Cloning Kit was purchased from New England Biolabs (Hitchin, UK); ROTI^®^PolTaqSMix was purchased from Carl Roth GmbH + Co. KG (Karlsruhe, Germany). The GeneJET PCR Purification Kit was purchased from Thermo Scientific™ (Waltham, MA, USA); the protein Ni-NTA Purification System K95001 from Thermo Fisher scientific (Altrincham, UK); and the Prestained Protein Ladder 10–180 kDa from Thermo ScientificTM PageRulerTM (Boston, MA, USA). Protein markers 10–310 kDa (ROTI^®^Mark TRICOLOR) were purchased from Carl Roth (Karlsruhe, Germany). DNA ladder 1 kb was purchased from Solis BioDyne (Tartu, Estonia). Isopropyl β-D-1-thiogalactopyranoside (IPTG) and N-lauroylsarcosine sodium salt were provided by Sigma-Aldrich (St. Louis, MO, USA).

### 2.2. Cloning of D-Carbamoylase Gene

The *Pseudomonas* sp. strain KNK003A is characterized by stable N-carbamoyl-D-amino acid amidohydrolase at 45 °C [11]. We amplified the gene coding for D-carbamoylase from this strain (GenBank: BAD00008) by PCR using the forward primer F (5′-GGAGATATACATATGACCCGTATTGTTAACGCTGC-3′) and the reverse primer R (5′-GTGGTGGTGCTCGAGCTGCGGCGGCGGCACC-3′). The reverse primer lacked the terminator motif, allowing the creation of the His-tagged gene variant. The primer pair contained motifs for the target gene and vector, along with respective restriction enzyme sites, to facilitate the subsequent gene cloning by the Gibson assembly [27].

The region of homology with the D-carbamoylase gene of *Pseudomonas* sp. strain KNK003A is underlined in both of the primers used. DNA amplification was carried out by PCR [28] with initial denaturation at 95 °C for 2 min, 30 cycles of denaturation at 95 °C for 30 s, annealing at 57 °C for 30 s, and elongation at 72 °C for 30 s. The purified PCR product was cloned according to protocol [27] into the expression vector pET24a(+), cleaved at the *Nde*I and *Xho*I restriction sites. The constructed plasmid pET24a(+)-*PsDcase* was transferred into *E. coli* strain BL21 Star by selecting kanamycin-resistant colonies on Luria-Bertani (LB) agar. The presence of the recombinant plasmid was confirmed using PCR analysis of transformed colonies (Appendix A) [29]. The recombinant *E. coli* BL21 Star/pET24a(+)-*PsDcase* cells were grown until OD_600_ 0.5, and D-carbamoylase synthesis was induced with 0.5–1 mM IPTG at 30 °C followed by incubation for a further 6–20 h.

### 2.3. Purification of Recombinant D-Carbamoylase

The cells were collected by centrifugation at 8000× *g* for 20 min, suspended in 100 mM phosphate buffer (pH 7.0) containing 1 mM DTT, and sonicated on ice for 5 min with 30 s on/off cycles at 25 kHz (Cole-Parmer^®^ 500-Watt Ultrasonic Processor, Vernon Hills, IL, USA). The lysate was cleared by centrifugation at 35,000× *g* for 30 min at 4 °C and additionally filtered through a syringe filter with a pore size of 0.45 μm. The resulting supernatant, corresponding to the soluble protein fraction of the crude extract, was analyzed by SDS-PAGE and further used to measure D-carbamoylase activity. Polyhistidine-labeled D-carbamoylase was purified using the Ni-NTA purification system and tested for enzymatic activity. The purity of the protein preparations was estimated using GelAnalyzer 23.1.1 by analyzing the respective traces from SDS-PAGE.

### 2.4. D-Carbamoylase Activity Assay

D-carbamolase activity was measured in the soluble fraction of the crude extract of purified protein in 100 mM phosphate buffer (pH 7.0) containing 1 mM DTT at 45 °C. The reactions were started by adding 0.1 mL of the appropriate substrate consisting of 100 mM N-carbamoyl-D-amino acid or 200 mM N-carbamoyl-DL-amino acid in 100 mM phosphate buffer (pH 7.0), after preincubating the reaction solutions at 45 °C for 10 min. After 15–120 min incubation, the reaction was stopped by adding 30% trichloroacetic acid (TCA) to a final concentration of 3% *w*/*v*. D-carbamoylase activity was determined using the orthophthalaldehyde (OPA) reagent [30]. Specifically, 3 mL of freshly prepared OPA reagent (0.1 M borate buffer, pH 9.6, 2.5 mM OPA, and 2.5 mM β-mercaptoethanol) was added to each sample, followed by incubation at 20 °C for 30 min. The D-amino acid concentrations were determined spectrophotometrically (Thermo Scientific Genesys 50 UV-vis spectrophotometer, US) based on the absorbance of the OPA reagent at 340 nm. Enzyme activity was expressed in units (U), with 1 U defined as the conversion of 1 μmol of substrate per minute. The specific activity of the enzyme was expressed in U/mg protein. All measurements were performed in at least three separate experiments with duplicates.

### 2.5. Solving of Inclusion Bodies

Significant amounts of D-carbamoylase were extracted from the inclusion bodies (IBs) using the modified method described by Peternel et al. [31]. The insoluble protein fraction IBs of the crude extract were separated by centrifugation at 10,000× *g* for 20 min at 4 °C. To dissolve the insoluble fraction, 80 μL of 0.2% N-lauroylsarcosine sodium salt solution prepared in 100 mM phosphate buffer (pH 7.0) was added to each 1 mg of wet sediment. The resulting mixture was incubated for 24 h at 20 °C with shaking at 100 rpm and centrifuged for 15 min at 4400× *g*. The resulting supernatant, corresponding to the solubilized target protein, was analyzed by SDS-PAGE and tested for D-carbamoylase activity.

### 2.6. Protein Assays

The D-carbamoylase expression levels were analyzed by SDS-PAGE, and the gels were stained with Coomassie Brilliant Blue R-250 to visualize protein bands. Cell-free extract was prepared by sonicating the cells and, after centrifugation, the supernatant was collected as the soluble fraction. The pellet remaining from this sample was resuspended in 0.5 mL of a buffer solution consisting of 50 mM Tris-Cl (pH 7.2), 5% glycerol, 5% SDS, and β-mercaptoethanol. The suspended solution was centrifuged again, and the supernatant was considered the insoluble fraction. The protein concentration was determined using the described methods of Groves and Davis, and Bradford [32].

The UniProt database was used to search purified D-carbamoylase against the available D-carbamylases downloaded from GeneBank. The alignment tools T-coffee [33] and MView [34] were used to align D-carbamoylase sequences. Homology modeling was performed using the SWISS-MODEL web server [35].

### 2.7. Molecular Docking

Ligand structures were generated using ChemBioDraw Ultra 12.0 or downloaded from the PubChem database. Energy minimization of ligand structures was carried out in OpenBabel [36] using the MMFF94 force field. For molecular docking, we used the Gnina 1.0 program, which uses a blind, whole docking approach [37]. The binding site was specified by providing a ligand file (autobox_ligand). The explicit random seed was set to 0 and exhaustiveness to 8. The convolutional neural net was set to none. Discovery Studio 2021 software was used to visualize protein–ligand interactions.

### 2.8. Molecular Dynamics

Molecular dynamics (MD) simulations were performed using the GROMACS 2023 simulation package [38]. The ligand molecule topology files were generated by the online server Swissparam [39]. The water chamber was created at a distance of at least 10 Å from the complex using the Tip3p water model and applying periodic boundary conditions in a dodecahedron simulation cell. To neutralize the system, Na^+^ and Cl^-^ ions were added to the system. MD was conducted to verify the quality of the model structure by testing its stability through 100 ps simulations at a constant temperature of 300 K (NVT-constant Number of particles (N), Volume (V), and Temperature (T)). Then, NPT (constant Number of particles (N), Pressure (P) and Temperature (T) optimization was carried out for 100 ps. MD was performed for 100 ns using a Charm27 force field. The gmx MMPBSA tool, based on AMBER’s MMPBSA.py [40], was used to perform the final free energy calculations.

### 2.9. English Language Correction

ChatGPT was used for correcting English language in some sentences.

## 3. Results

### 3.1. Gene Cloning, D-Carbamoylase Expression and Purification

The carbamoylase gene of *Pseudomonas* sp. strain KNK003A was amplified by PCR, and the purified PCR product was cloned into the expression vector pET24a(+) to obtain the recombinant plasmid pET24a(+)-*PsDcase*. This plasmid was transferred into *E. coli* BL21 Star cells, and the presence of the recombinant plasmid was confirmed by direct colony PCR analysis [29].

To overexpress D-carbamoylase, individual recombinant colonies of *E. coli* BL21 Star/pET24a(+)-*PsDcase* cells were induced with 0.75 mM IPTG at 30 °C, followed by incubation for 6, 8, and 20 h. The centrifuged cells were disrupted by sonication, the crude extract was separated by centrifugation, and the supernatant and precipitated fractions were subjected to SDS-PAGE analysis, which showed a high expression of the target protein (Figure 2A). However, only a small portion of the D-carbamoylase produced was present in the soluble fraction, while a significant portion remained trapped in the IBs.

Protein aggregation occurs regardless of the bacterial species [41,42]. In bacteria in which recombinant proteins are not secreted into the medium, the aggregation of recombinant proteins leads to the formation of IBs [43]. Over the past decades, several studies have reported that it is often difficult to achieve overexpression of soluble carbamoylase in heterologous bacterial expression systems, especially in *E. coli* [44,45,46]. Poorly soluble functional proteins can be extracted from insoluble protein aggregates using mild detergents [47]. N-lauroylsarcosine sodium is one of the most commonly used agents for the extraction of nondenaturing protein from IBs. We extracted a significant amount of the target protein from IBs using 0.2% N-lauroylsarcosine sodium solution, but the solubilized protein D-carbamoylase had low biological activity.

Another possible way to solve the problem of insoluble protein is to reduce the expression level of the target protein to prevent the formation of insoluble aggregates. The use of low temperatures to prevent the formation of IBs was first demonstrated for human proteins, interferon α2, and human interferon γ [48]. We induced the production of our D-carbamoylase at a lower temperature; this did not produce the expected result at 25 °C, and D-cabamoylase still remained aggregated. The absence of the effect of solubilization of recombinant proteins due to a decrease in cell growth temperature was also observed during the purification of other proteins [45,49].

Finally, the inability to obtain a soluble and active enzyme using two different approaches was partially overcome by purifying D-carbamoylase using the Ni-NTA protocol PageRuler^TM^ (Thermo Scientific^TM^), which resulted in a fairly pure and active enzyme despite its loss during purification (Figure 2B).

### 3.2. The Search of Possible Carbamoylases

We identified 32 proteins corresponding to the term “D-carbamoylase” in the UniProt database. The study of the properties of the proteins suggested that D-carbamoylase of *Ensifer adhaerens* S-5 and *Pseudomonas* sp. strain KNK003A may be suitable enzymes in terms of compatibility with D-hydantoinase of the thermophilic bacterium *Geobacillus stearothermophilus*, which we studied [26]. In particular, the enzyme’s ability to open bulky N-carbamoylamino acids, such as N-carbamoyl-D-phenylalanine, N-carbamoyl-p-hydroxy-D-phenylglycine, and N-carbamoyl-D-phenylglycine, was considered an advantage.

Multiple sequence analysis of the mentioned enzymes with *Agrobacterium* sp. KNK712 (GenBank: P60327) and *Arthrobacter crystallopoietes* DSM 20117 (GenBank: AAO24770.1) carbamoylases revealed highly conserved regions (Appendix A). High similarity was observed between four aligned sequences, especially in the regions responsible for the catalytic activity of the enzyme. It is important to note that the catalytic triad Cys-Glu-Lys [22,23] is presented as Cys172, Glu47, and Lys127 in *Pseudomonas* sp. strain KNK003A D-carbamoylase.

### 3.3. Molecular Docking of D-Carbamoylase

To understand the interactions between the ligands and enzyme of interest, we generated a model of *Pseudomonas* sp. strain KNK003A D-carbamoylase monomer. Mutant D-carbamoylase from *N. indicus* C115 (PDB code: 6le2, resolution 2.14 Å) was used as a template for the homology model of the *Pseudomonas* sp. strain KNK003A [11] D-carbamoylase model due to its having the highest sequence similarity (0.48), Seq Id (60.33), and coverage (0.98) (see Appendix A).

Molecular docking of D-carbamoylase was used to detect the binding pocket of the enzyme and the correct orientation of the ligand in the pocket. The *Pseudomonas* sp. strain KNK003A enzyme model suggested the possibility of exhibiting activity towards N-carbamoyl-D-tryptophan (−5.1 kcal/mol) (Figure 3A). To select the correct position of N-carbamoyl-D-tryptophan in the binding pocket, we considered the protein conformation with the lowest binding energy and ligand orientation. From the various positions proposed by the docking algorithm, a position was selected that allowed the formation of hydrogen bonds of the amino acids Arg176 and Asn173 with the exposed carbamoyl group of the ligand, which is necessary for catalysis. The chosen position allowed the target group of the ligand to interact closely with the catalytically important amino acids Lys127 and Cys172. N-carbamoyl-para-hydroxy-D-phenylglycine (−4.9 kcal/mol), N-carbamoyl-D-valine (−4.19 kcal/mol), N-carbamoyl-D-methionine (−2.8 kcal/mol), N-carbamoyl-D-phenylglycine (−3.8 kcal/mol), N-carbamoyl-D-phenylalanine (−4.2 kcal/mol), N-carbamoyl-D-leucine (−3.7 kcal/mol), N-carbamoyl-D-alanine (−4.4 kcal/mol), N-carbamoyl-beta-alanine (−2.6 kcal/mol), and N-carbamoyl-D-glycine (−3.1 kcal/mol) are among the ligands that have affinity for D-carbamoylase and can potentially be degraded by the enzyme (the affinities presented are not those with the lowest energy, but those best fitting the binding pocket). The docking image in Figure 3B shows that D-carbamoylase has an affinity for bulky substrates, indicating that there is enough space in the binding pocket to accommodate relatively large substrates. In addition, given the hydrophobic structure of N-carbamoyl-D-tryptophan and other ligands, we checked the hydrophobic nature of the pocket, which is represented in Figure 3B. The binding of N-carbamoyl-D-tryptophan to D-carbamoylase was previously described for D-hydantoinase [26], which is important for harmonizing the catalytic potential of both enzymes in the “hydantoinase process”. Thus, molecular docking showed that the modeled D-carbamoylase had significant affinity for a wide range of substrates. Further modeling studies were continued using N-carbamoyl-D-tryptophan as a substrate.

### 3.4. Molecular Dynamics of D-Carbamoylase Binding to Carbamoylated Amino Acids

To confirm the role of putative amino acids in the substrate binding and D-carbamolase catalysis, 100 ns molecular dynamics simulations of the interaction of the *Pseudomonas* sp. strain KNK003A enzyme alone and in complex with N-carbamoyl-D-tryptophan were performed. To assess the deviation from the primary structure and stability of the protein and complex, the root mean square deviation (RMSD) was monitored. As shown in Figure 4A, stabilization of a complex protein–ligand is achieved after ~50 ns. The average RMSD for the protein alone was calculated to be 0.26 nm and for the complex to be 0.33 nm. From the RMSD analysis, it can be concluded that the molecular dynamics trajectories are stable throughout the simulation and within the acceptable range and that the data can be used for further analysis.

To measure the fluctuations of each amino acid, root mean square fluctuations (RMSF) were also assessed (Figure 4B). The analysis showed that the D-carbamoylase amino acid fluctuations for both models remained within the acceptable range throughout the 100 ns simulation. The average RMSF value for the protein was 0.136 ± 0.02 nm, and for the complex, it was 0.139 ± 0.03 nm. The initial N-terminal and final C-terminal residues in the curves showed higher flexibility compared to the rest of the protein sequence. For the amino acids (Lys127, Gly132, His144, Cys172, Asn173, Arg176, Pro199) interacting with the ligand, the RMSF values in the case of the protein–ligand complex were lower (except Arg176) compared to the RMSF of protein, indicating the stabilizing effect of the ligand (Figure 4C). Thus, the D-carbamoylase binding site RMSF exhibits lower flexibility when binding to N-carbamoyl-D-tryptophan, indicating stability of catalytic activity.

To evaluate the structural adaptation of the ligand–protein complex caused by the ligand we evaluated the radius of gyration (Rg) for both the protein and the complex. As can be seen in Figure 5A, the process of adaptation continues till 35–40 ns; then, the structures become almost convergent.

The average Rg value for the protein is estimated to be 1.89 ± 0.05 nm and for the complex 1.91 ± 0.05 nm. Thus, both models showed relatively similar folding behavior, but with slightly more flexibility for the protein–ligand complex. To analyze the interaction of the complex and the protein with the solvent, solvent accessible surface area (SASA) was calculated during 100 ns simulations (Figure 5B). The average SASA value for a protein was estimated to be 147.3 ± 1.98 nm^2^, and for the protein–ligand complex, it was estimated to be 149.7 ± 2.04 nm^2^. Thus, the SASA values are close for both models.

The MD simulations showed that carbamoyl-D-tryptophan forms hydrogen bonds with the enzyme D-carbamoylase. Based on the curve of the number of hydrogen bonds over a 100 ns simulation time, carbamoyl-D-tryptophan forms up to 11 hydrogen bonds with the protein per nanosecond (Figure 6A; see the maximum number on the y-axis), 6 hydrogen bonds with Arg176 (Figure 6B), and 4 hydrogen bonds with Asn173 of the protein per nanosecond (Figure 6C). NH_2_ groups of Arg176 form hydrogen bonds with the oxygen atom of the carbonyl group (C=O) of the ligand, and arginine acts as a donor. In the interaction between the carbonyl group of Asn173 and the NH group of N-carbamoyl-D-tryptophan, asparagine’s carbonyl group (C=O) acts as a hydrogen bond acceptor. With four amino acid residues that form hydrogen bonds, D-carbamoylase has two polar amino acids (Asn, Arg) and two non-polar amino acids (Gly, Pro). Polar amino acids enhance binding affinity through hydrogen bonding with polar groups on the ligand or through strong electrostatic interactions with charged or polar groups on the ligand. Non-polar amino acids provide flexibility (Gly) and influence the structure of the binding site due to their rigid cyclic structure (Pro). The formation of these bonds suggests a stable interaction between the ligand and the protein and confirms the results of the molecular docking.

The molecular dynamics of hydrogen bond formation during the interaction of D-carbamoylase with N-carbamoyl-D-tryptophan during 100-nanosecond simulations may be used as an indicator of the appropriate state of the binding site during catalysis.

### 3.5. Calculation of Molecular Mechanics of Poisson–Boltzmann Surface Area (Mmpbsa)

Binding free energy calculations were performed using the gmx_mmpbsa tool to calculate the average binding free energy with an MD trajectory of 100 nanoseconds.

The ΔG value is calculated based on the following equation:ΔG = [E_complex_ − (E_receptor_ + E_ligand_)] + [ΔG_PB_ + ΔG_SA_](1)
where E_complex_, E_receptor_, E_ligand_ are the molecular mechanics energies of the complex, and ΔG_PB_, ΔG_SA_ are the changes in solvation free energies.

Van der Waals interactions (ΔVDWAALS = −20.83 ± 1.54 kcal/mol), electrostatic interactions (ΔEEL = 49.51 ± 5.71 kcal/mol), and the non-polar contribution of repulsive solute–solvent interactions to the solvation energy (ΔENPOLAR = −2.62 ± 0.25) kcal/mol) make the main contribution to the formation of the enzyme–ligand complex, while non-polar interactions in the solvation system (ΔEPB = 59.63 ± 7.01 kcal/mol) have a negative impact on the generation of binding energy. ΔGGAS and ΔGSOLV were also included in the total binding free energy. The total binding energy of the D-carbamoylase/N-carbamoyl-D-tryptophan complex was −13.32 ± 1.57 kcal/mol.

Although the large contribution of van der Waals interactions to the total binding energy suggests a hydrophobic contact of the ligand with the receptor, predominantly conditioned by the hydrophobic amino acids (Phe53, Pro131, Gly132 and Pro199) located in the binding site (Figure 3A), the surface shown in Figure 3B is dominated by blue, suggesting low hydrophobicity.

### 3.6. D-Carbamoylase Substrate Specificity

Purified D-carbamoylase was analyzed for enzymatic activity based on the obtained molecular docking data. The highest levels of enzyme activity towards various substrates were detected at 45 °C with incubation for 120 min (Table 1), using IPTG-induced D-carbamoylase incubated at 30 °C with shaking for 20 h.

As expected, D-carbamoylase showed strong specificity for N-carbamoyl-D-amino acids and was not active against the tested N-carbamoyl-L-amino acids. Low specificity was observed for N-carbamoylated DL-amino acids but was less reflective of the expected two-fold difference between the D- and L-amino acids. The enzyme D-carbamoylase showed activity against aliphatic and aromatic N-carbamoylamino acids in vitro, which was in good agreement with the in silico data. Additionally, higher activity was observed against bulky substrates, such as N-carbamoyl-D-tryptophan, N-carbamoyl-D-phenylalanine, N-carbamoyl-D-valine, and N-carbamoyl-D-leucine.

D-carbamoylase belongs to a superfamily of amidohydrolases that catalyze the breakdown of a wide range of N-carbamoyl-D-amino acids. Both in vitro and in silico methods have been used to characterize the structure and behavior of the bacterial enzyme. Molecular docking revealed the affinity of D-carbamoylase for various substrates, and molecular dynamics studies showed stability and the constant formation of hydrogen bonds, especially with N-carbamoyl-D-tryptophan, indicating the involvement of the substrate binding site of the enzyme. These data are encouraging indicators for the subsequent production of optically pure D-amino acids.

Our data suggest that the activity of D-carbamoylase can still be increased by site-directed mutagenesis to produce more D-amino acids from the corresponding precursors. Moreover, docking analysis revealed the molecular basis for the increased affinity of D-carbamoylase for bulky N-carbamoyl-D-tryptophan, which may be important for harmonizing its catalytic potential with that of D-hydantoinase in the overall two-enzyme process. Thus, enzymatic and docking analyses give a positive assessment of the two bacterial enzymes, D-hydantoinase, as described previously [26], and D-carbamoylase, described in this work, for the implementation of the coupled biotechnological process. D-carbamoylase and hydantoinase are attractive catalytic enzymes for the biotechnological production of pharmaceuticals.

## 4. Discussion

The traditional molecular cloning of a gene encoding a desired protein typically requires knowledge of the nucleotide sequence at the beginning and end of the gene. Advances in the chemical synthesis of longer DNA sequences have accelerated the cloning of an entire gene when the desired bacterium is unavailable. Here, we chemically synthesized a 939 bp gene encoding D-carbamoylase from *Pseudomonas* and cloned it into *E. coli*.

Biochemical analysis established the identity of the D-carbamoylase activity with the originally described enzyme [11] and also provided additional information on the cleavage spectrum of the new D-carbamoyl derivatives of amino acids. Importantly, the ability of the enzyme to cleave large N-carbamoyl-D-amino acids such as N-carbamoyl-D-tryptophan or N-carbamoyl-D-phenylalanine indicates a real advantage of biotechnological production of D-amino acids over chemical synthesis. *Pseudomonas* D-carbamoylase exhibits poor solubility in *E. coli* cells, which can be improved to some extent by treatment with sodium N-lauroylsarcosine or by expression of the enzyme in cells grown at 25 °C. However, after testing both methods, we preferred the Thermo ScientificTM method using PageRuler^TM^ to purify the His-tagged protein, which allowed us to obtain sufficiently pure and active D-carbamoylase for further study of the enzymatic activity of the protein. It is important to note that the two-step “hydantoin process” does not include an enzyme purification step.

Detailed virtual analysis of the cloned D-carbamoylase protein provides valuable information about the structural features of the enzyme. Molecular docking revealed the affinity of D-carbamoylase for bulky N-carbamoyl-D-amino acids, which provides sufficient space in the binding pocket to accommodate other bulky substrates. The molecular dynamics of D-carbamoylase binding to N-carbamoyl-D-tryptophan shows rapid, nearly 50 ns, stabilization of the protein–ligand complex. This property is important for the two-step synthesis of large structures of D-tryptophan and other D-amino acids by combining the catalytic potential of D-hydantoinase [26] and D-carbamoylase into a single process.

## 5. Conclusions

In contrast to chemical synthesis, the two-step biotechnological production of D-amino acids has technological advantages, as the process can be environmentally friendly, recyclable, and safe for operating personnel. The enzymes D-hydantoinase and D-carbamoylase, which are involved in various biological processes in bacterial hosts, are promising catalysts for the co-synthesis of D-amino acids and the possible production of pharmaceuticals, such as semi-synthetic antibiotics and peptide hormones, as well as pesticides for agronomy. Our in vitro and in vivo data on D-hydantoinase, as described previously [25,26], and on D-carbamoylase, as described in the present study, allow us to foresee the possibility of establishing a harmonized production of pure D-amino acids by the combined action of both enzymes.

## Figures and Tables

**Figure 1 biotech-13-00040-f001:**
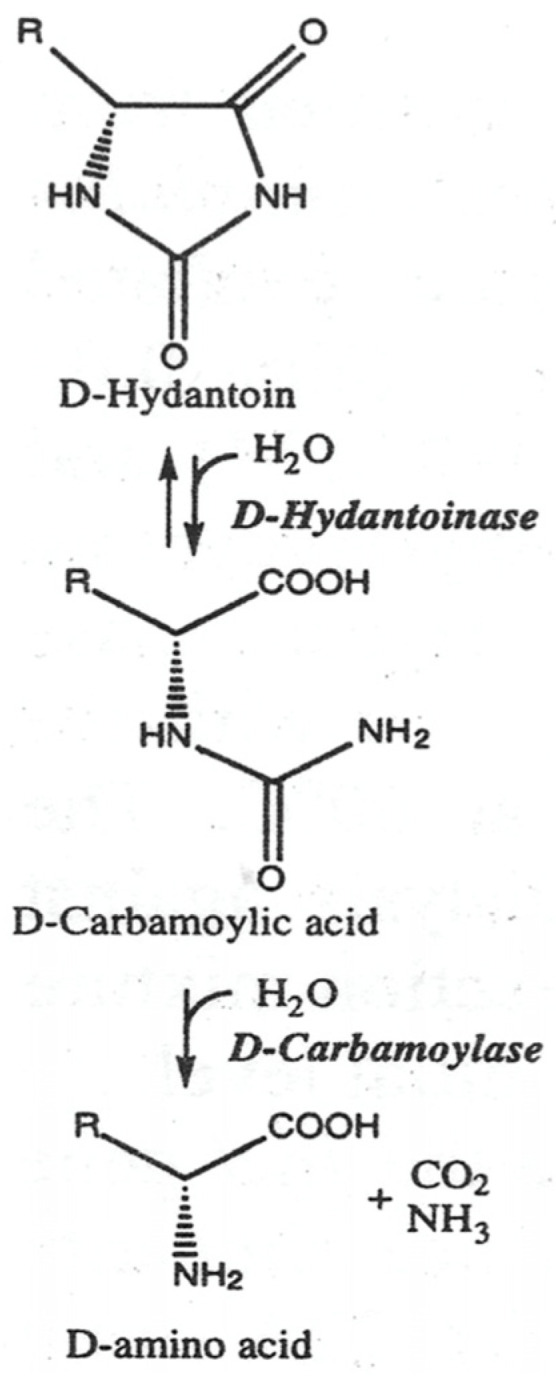
Two-step “hydantoinase process” for production of D-amino acids.

**Figure 2 biotech-13-00040-f002:**
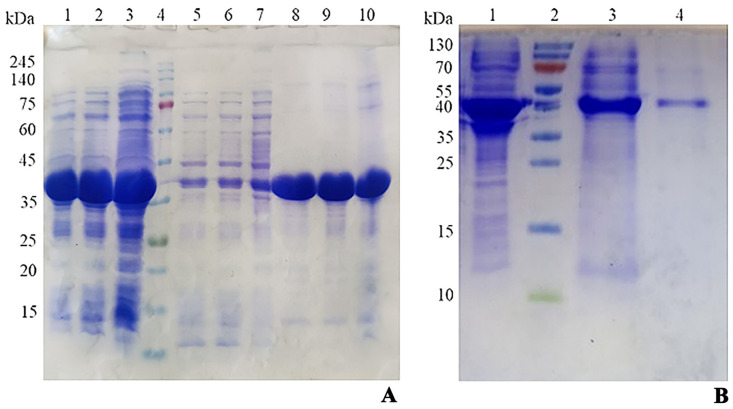
(**A**) SDS-PAGE of recombinant *E. coli* BL21 Star/pET24a(+)-*PsDcase* cells. Lines 1–3 represent proteins induced for 6, 8, and 20 h in crude extract; line 4—protein markers (ROTI^®^Mark TRICOLOR); lines 5–7—proteins of the supernatant fraction of the same samples; lines 8–10—proteins in the precipitated insoluble fraction of the same samples. (**B**) SDS-PAGE of D-carbamoylase purified with Ni-NTA protocol. Line 1—crude extract induced for 20 h; line 2—protein markers (Thermo Scientific^TM^ PageRuler^TM^); line 3—partially purified protein; line 4—purified protein (93.6%).

**Figure 3 biotech-13-00040-f003:**
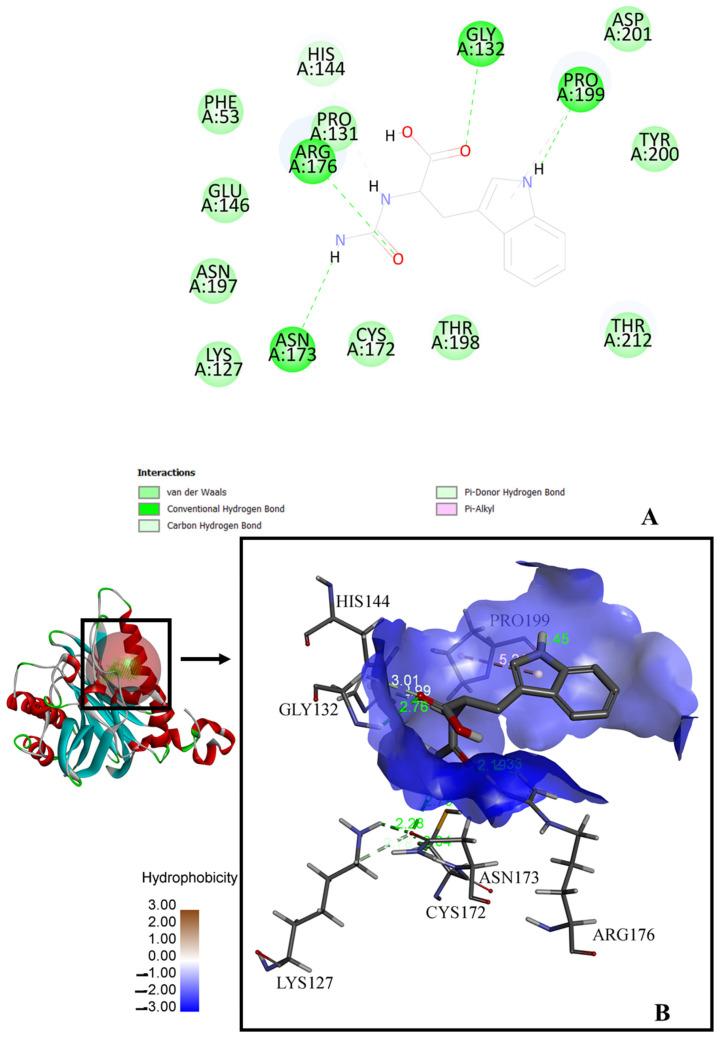
Docking model of the interaction of *Pseudomonas* sp. KNK003A D-carbamoylase with N-carbamoyl-D-tryptophan in the substrate-binding pocket. (**A**) Two-dimensional image of molecular interactions in the enzyme. (**B**) Three-dimensional image of molecular interactions in the enzyme. Structure with green and purple dotted lines shows hydrogen bonds and hydrophobic interactions and their lengths in angstroms, respectively. The surface indicates that the binding pocket is mainly hydrophilic along with hydrophobic regions.

**Figure 4 biotech-13-00040-f004:**
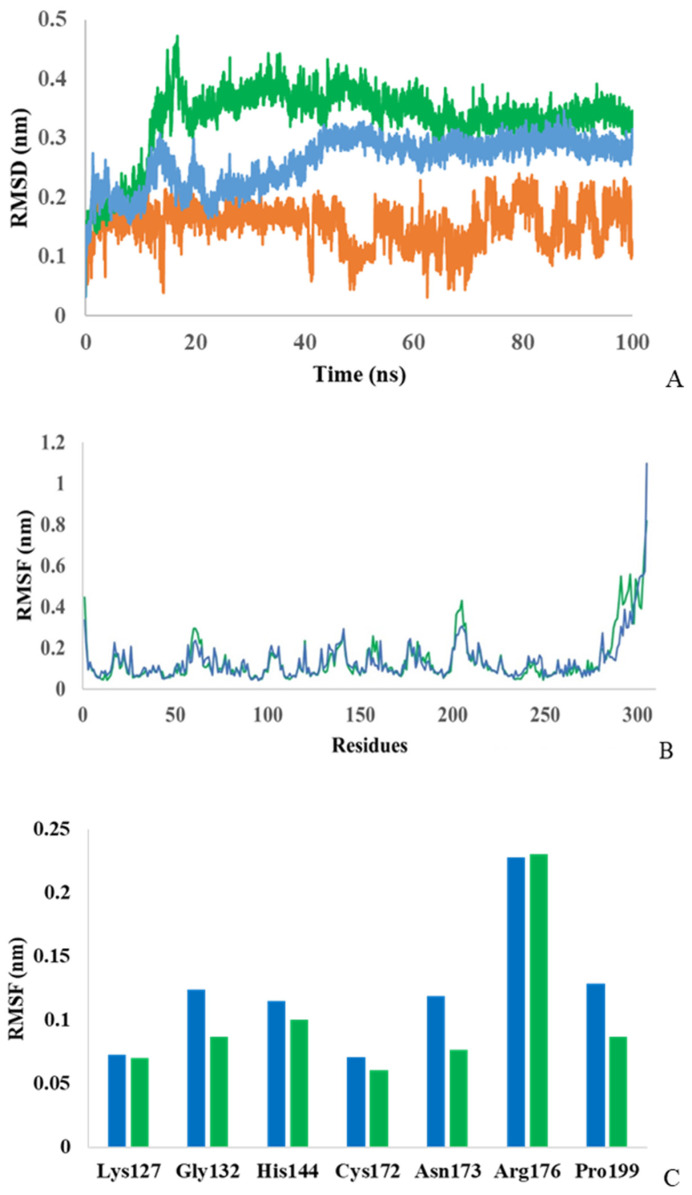
Monitoring RMSD (**A**) for D-carbamoylase alone (blue), for the D-carbamoylase/N-carbamoyl-D-tryptophan complex (green), and for N-carbamoyl-D-tryptophan (orange) and RMSF (**B**) for D-carbamoylase alone (blue) and for the complex (green), as well as RMSF (**C**) of catalytically important amino acids over a 100 ns simulation.

**Figure 5 biotech-13-00040-f005:**
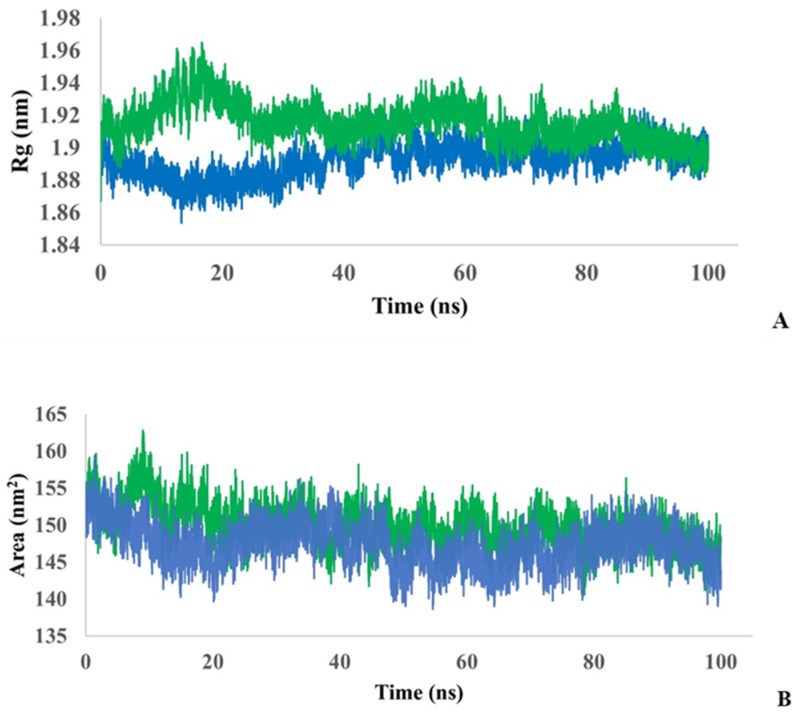
Monitoring of the radius of gyration (**A**) and monitoring of the SASA (**B**) in protein (blue) and protein–ligand complex (green) during 100 ns simulation.

**Figure 6 biotech-13-00040-f006:**
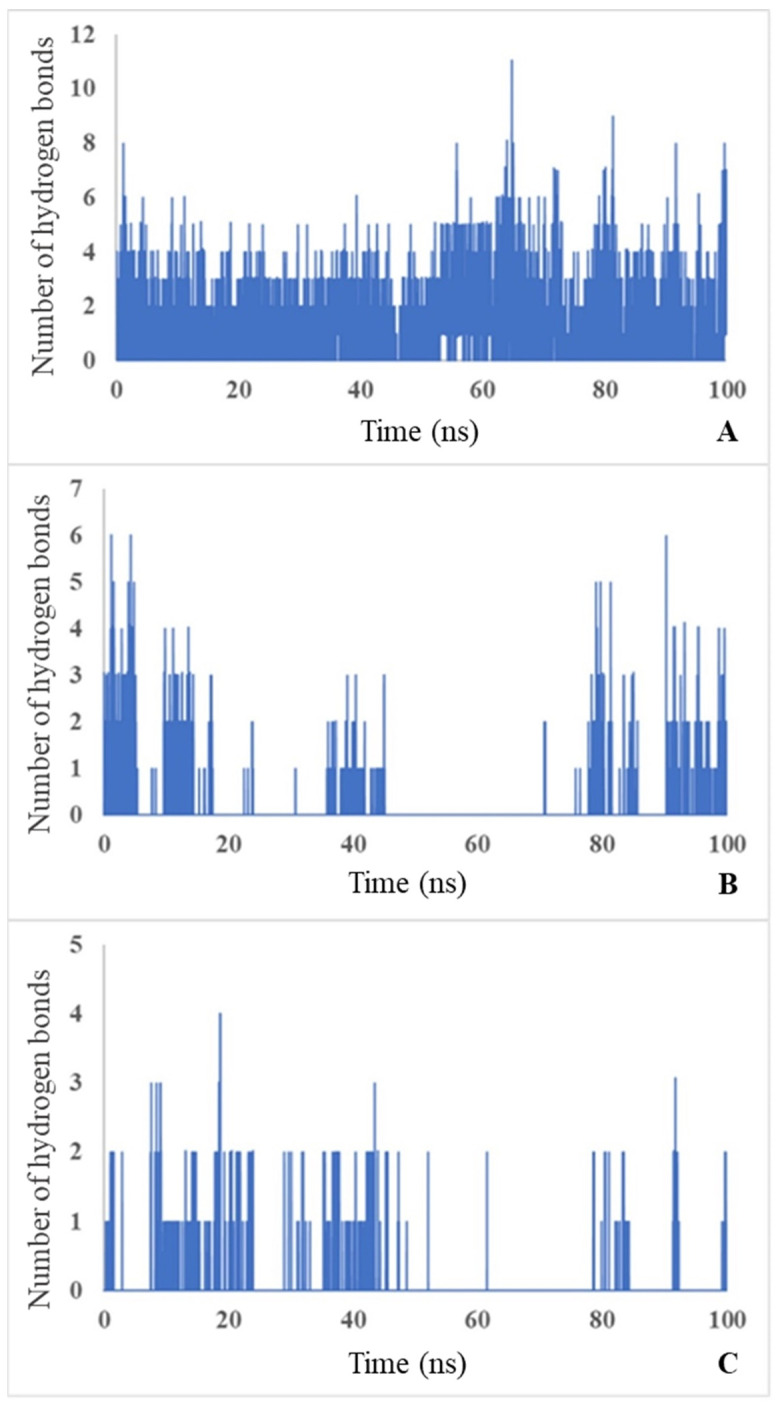
Number of H bonds throughout the simulation time of 100 ns between protein and ligand (**A**), Arg176 and ligand (**B**), Asn173 and ligand (**C**). Blue bars indicate the number of bonds formed during each ns.

**Table 1 biotech-13-00040-t001:** Substrate specificity of D-carbamoylase to N-carbamoyl-D and N-carbamoyl-L amino acids.

Compound	Specific Activity (U/mg)
N-carbamoyl-D-tryptophan	0.449 ± 0.03
N-carbamoyl-D-phenylalanine	0.265 ± 0.02
N-carbamoyl-D-alanine	0.461 ± 0.04
N-carbamoyl-DL-alanine	0.368 ± 0.03
N-carbamoyl-D-valine	0.442 ± 0.04
N-carbamoyl-D-leucine	0.190 ± 0.02
N-carbamoyl-L-tryptophan	0
N-carbamoyl-L-leucine	0

## Data Availability

The original contributions presented in the study are included in the article/Appendix A, further inquiries can be directed to the corresponding authors.

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
