# Peer review of "Structural Analysis and Substrate Specificity of D-Carbamoylase from Pseudomonas"

_biotech, 2024, doi:10.3390/biotech13040040_

Round 1
Reviewer 1 Report
Comments and Suggestions for Authors
The manuscript provides, interest among the readers, however the author needs to clarify certain queries regarding the outcome of the manuscript.
regrading the molecular docking studies, the authors have not provided the binding energy and other energy parameters, interestingly, there appears several conventional and non conventional hydrogen-bonding interactions for N-carbamoyl-D-tryptophan. the question is that how many conformers were generated? what was the difference in binding energy between the stable and least stable conformers and the nature of the bimolecular interactions.
what is the nature of docking and site: whether the ligands bind at Pseudomonas sp. KNK003A D-carbamoylase , example at turns/coils/sheets/ etc.. a discussion is required regarding these studies.
the authors have restricted docking studier to only N-carbamoyl-D-tryptophan., however they have carried out PMR and C13 NMR to many N-carbamoyl derivatives, what is the reason that they have not involved other derivatives in their theoritical studies.
docking and simulation studies references are not well elucidated.. have the authors accounted for Lipinski's rule of five?
what is the role of polar versus non-polar amino acids here on the interaction affinity. the authors should provide more information
Comments on the Quality of English Language
nil
Author Response
Comments and Suggestions for Authors
The manuscript provides, interest among the readers, however the author needs to clarify certain queries regarding the outcome of the manuscript.
Comment 1 [regrading the molecular docking studies, the authors have not provided the binding energy and other energy parameters, interestingly, there appears several conventional and non conventional hydrogen-bonding interactions for N-carbamoyl-D-tryptophan. the question is that how many conformers were generated? what was the difference in binding energy between the stable and least stable conformers and the nature of the bimolecular interactions.]
Response 1 [Edited in the text (lines 275, 282-288). We did not generate conformers. We downloaded the structure from an online database, obtained an energy-minimized structure, and used it in molecular docking.]
Comment 2 [what is the nature of docking and site: whether the ligands bind at Pseudomonas sp. KNK003A D-carbamoylase , example at turns/coils/sheets/ etc.. a discussion is required regarding these studies.]
Response 2 [Asn173 and Arg176 are predicted to be in the alpha helix, and Gly132, His144, and Pro199 are predicted to be in the turns/loops.]
Comment 3 [the authors have restricted docking studier to only N-carbamoyl-D-tryptophan., however they have carried out PMR and C13 NMR to many N-carbamoyl derivatives, what is the reason that they have not involved other derivatives in their theoritical studies.]
Response 3 [The provided information on PMR and C13 NMR was necessary to assess the purity of the synthesized substrates for further use in biochemical analysis.]
Comment 4 [docking and simulation studies references are not well elucidated. have the authors accounted for Lipinski's rule of five? ]
Response 4 [Lipinski's rule of five on the medicinal properties of N-carbamoyl-D-tryptophan is as follows: mass 234.00; hydrogen bond donor: 0; hydrogen bond acceptors: 6; LOGP is -2.125320; Molar Refractivity 48.828499. We think that the information on Lipinski rule should not be included in the manuscript text.]Comment 5 [what is the role of polar versus non-polar amino acids here on the interaction affinity. the authors should provide more information]
Response 5 [Edited in the text (lines 354-370).]Thank you very much for reviewing our manuscript.
Comments on the Quality of English Language
nil
Submission Date 23 August 2024
Date of this review 30 Aug 2024 17:50:33
Reviewer 2 Report
Comments and Suggestions for Authors
The manuscript entitled “Structural analysis and substrate specificity of D-carbamoylase from Pseudomonas” submitted by Paronyan M et al., has been reviewed and the comments are as follows:
Major comments:
Line 226: Lowering the temperature to 25°C did not result in soluble protein production. Might try at a temperature lower than 20°C during post-induction and incubation for a longer period (ex: 10 hrs) might produce a higher quantity of soluble proteins.
Line 232: The D-carbamoylase protein was purified by compromising the loss during the process reported (Fig. 2B, lane 4 has mild visible bands at different regions, which may be due to a de-staining issue). However, the assessment to measure the purity of the protein is missing in the manuscript, which can be included to showcase the purity of the recombinant D-carbamoylase protein.
Line 241: The basis for the selection of 4 specific sequences from 32 hits obtained can be explained elaboratively since that serves as the basis for the following structural studies.
Line 247: How does the author conclude the statement as “Agrobacterium sp. KNK712 (GenBank: P60327) and Arthrobacter crystallopoietes DSM 20117 (GenBank: AAO24770.1) carbamoylases revealed highly conserved regions”. Do the authors attempt to perform the multiple sequence analysis of 32 hits and select these two hits (GenBank No: P60327, and AAO24770.1) among which are based on some criteria? Justify?
Provided the complete list of 32 hits and their associated details (organism name, GenBank No, Sequence length, etc), in the supplementary information.
What do the terms “screen” in Line 175 and “virtual screening” in Line 239? No such relevant details are seen in the manuscript.
Docking is performed, however docking score or the binding energy is not included, without which how the docking process is evaluated.
Represent the distance values in two-digit decimals (ex: 5.21 instead of 5.21137)
Line 304: Apo protein or native protein instead of Single protein
Line 118 to 120: Rephrase the entire region for better clarity to the readers.
Author Response
Comments and Suggestions for Authors
The manuscript entitled “Structural analysis and substrate specificity of D-carbamoylase from Pseudomonas” submitted by Paronyan M et al., has been reviewed and the comments are as follows:
Major comments:
Comment 1 [Line 226: Lowering the temperature to 25°C did not result in soluble protein production. Might try at a temperature lower than 20°C during post-induction and incubation for a longer period (ex: 10 hrs) might produce a higher quantity of soluble proteins.]
Answer 1 [We attempted to carry out the induction process at 25°C. However, at this temperature, the E. coli BL21 Star/pET24a(+)-PsDcase culture grew poorly and produced less biomass. In addition, the synthesized protein continued to form inclusion bodies. As a result, we did not use the induction process at lower temperatures.]
Comment 2 [Line 232: The D-carbamoylase protein was purified by compromising the loss during the process reported (Fig. 2B, lane 4 has mild visible bands at different regions, which may be due to a de-staining issue). However, the assessment to measure the purity of the protein is missing in the manuscript, which can be included to showcase the purity of the recombinant D-carbamoylase protein.]
Answer 2 [Edited in the text (lines 147-149, 244).
The purity of the protein preparations was estimated using GelAnalyzer 23.1.1 by analyzing therespective traces from SDS-PAGE. Specifically, for Fig. 2B, lane 4, a protein purity of 93.6% was estimated after removing from the trace the portion that did not penetrate into the separation gel.]
Comment 3 [Line 241: The basis for the selection of 4 specific sequences from 32 hits obtained can be explained elaboratively since that serves as the basis for the following structural studies.]
Answer 3 [The objects, with the exception of the Pseudomonas sp. KNK003A strain, were selected based on the substrate specificity exhibited by D-carbamoylase, which is approximately the same among these bacteria.]
Comment 4 [Line 247: How does the author conclude the statement as “Agrobacterium sp. KNK712 (GenBank: P60327) and Arthrobacter crystallopoietes DSM 20117 (GenBank: AAO24770.1) carbamoylases revealed highly conserved regions”. Do the authors attempt to perform the multiple sequence analysis of 32 hits and select these two hits (GenBank No: P60327, and AAO24770.1) among which are based on some criteria? Justify?]
Answer 4 [We performed multiple sequence analysis for only 4 proteins, as described in the Supplementary material.]
Comment 5 [Provided the complete list of 32 hits and their associated details (organism name, GenBank No, Sequence length, etc), in the supplementary information.]
Answer 5 [Since the strain Pseudomonas sp. KNK003A was chosen for further study, we provide information only on this bacterium.]
Comment 6 [What do the terms “screen” in Line 175 and “virtual screening” in Line 239? No such relevant details are seen in the manuscript.]
Answer 6 [Edited in the text (lines 184, 250).]
Comment 7 [Docking is performed, however docking score or the binding energy is not included, without which how the docking process is evaluated.]
Answer 7 [Edited in the text (lines 275, 282-288). Binding energy data have been added to the manuscript text. Distance values ​​have been truncated to two-digit decimal places (e.g. 5.21 instead of 5.21137) to improve the quality of the new image of the bound protein.]
Comment 8 [Line 304: Apo protein or native protein instead of Single protein]
Answer 8 [Edited in the text (line 349). The term “single” was removed.]
Comment 9 [Line 118 to 120: Rephrase the entire region for better clarity to the readers.]
Answer 9 [Edited in the text (lines 117-123).]
Thank you very much for reviewing our manuscript.
Submission Date 23 August 2024
Date of this review 11 Sep 2024 08:33:00
Reviewer 3 Report
Comments and Suggestions for Authors
Comments and Suggestions for Authors
biotech-3197981-peer-review-v1
’Structural analysis and substrate specificity of D-carbamoylase from Pseudomonas’
Introduction
Lines 70-72: In which databanks are data on 3-dimensional crystal structures of wild-type and mutant D-carbamoylases deposited?
Lines 84-86: „Structural analysis of the enzyme's three-dimensional structure provides important information useful for production of the desired D-amino acid from D-hydantoins.” In the presented work, such a structural analysis was not performed. The structure of the modeled protein was not even shown, the location of the putative binding site was not indicated, and the amino acids that form the binding site were not listed.
2.7. Molecular docking
Please provide values for the parameters used during docking.
2.8. Molecular dynamics
Please provide information on the boundary conditions applied during the MD simulations.
3.3. Molecular docking of D-carbamoylase
The work is missing a figure with the three-dimensional structure of the modeled protein, including the putative binding site location.
Near Figure 3 ... “the hydrophobic nature of the pocket can be assumed”. We do not have to assume that the binding site is hydrophobic. Discovery Studio Visualizer offers tools that can help determine the nature of the binding site, find hydrophobic amino acids, or show the surface of the binding site with mapped hydrophobicity.
Figure 3. Figure 3A should have a higher resolution. Figure 3B: The catalytically important amino acids Lys127 and Cys172, mentioned in line 270, should be shown in Figure 3B. Figure caption: The figure caption lacks an explanation of which color corresponds to which type of interaction – for example, we don’t know which are conventional hydrogen bonds and which are hydrophobic interactions. ‘Structure with dotted lines showing bonds and their lengths in angstroms’ suggests covalent bonds, whereas these are non-bonding intermolecular interactions, including but not limited to hydrogen bonds. Furthermore, covalent bond lengths and the distances over which intermolecular interactions occur are usually given to three decimal places, not five.
Since the activity of this enzyme has been experimentally tested on several compounds (3.6. D-carbamoylase substrate specificity), including six ‘D-substrates’, I would like to see information (figures similar to Figure 3) on what poses these ligands adopt in the active site and in what interactions with the protein they participate. I mean here, for example, information obtained in docking, although MD simulations would be even more valuable.
3.4. Molecular dynamics of D-carbamoylase binding to carbamoylated amino acids
The RMSD plot for the ligand (N-carbamoyl-D-tryptophan molecule) is missing – it should also stabilize in the active site.
How do the authors explain the differences in RMSF for the complex and the APO protein? Does ligand binding stabilize or destabilize the structure of the D-carbamoylase? Where in the protein chain are the amino acids with higher RMSF? In the binding site, outside it, are they loops or helices? There is no information about amino acids' numbers (indices) constituting the active site. For the reader, the information that "for amino acids representing the binding cavity, the RMSF values were lower than or close to the mean RMSF value" is impossible to verify. It would also be interesting to compare the RMSF of the binding site in the APO form and the complex.
Lines 295-296: ‘To understand whether the simulated protein stably folds or unfolds over the course of a 100 ns simulation’ – Formally, we are not dealing with protein folding or unfolding. The changes in Rg and RMSD up to around 35-40 ns indicate the structural adaptation of protein to the new situation when there is a ligand in the active site. However, from about 70 ns, the RMSD and Rg graphs of the APO form and the complex become almost convergent. Therefore, if the binding site is sufficiently compact, one can also analyze the course of Rg changes during the simulation for the active site (amino acids that form it), not only the entire protein.
‘1 ns’ in lines 313 and 314 suggests the first nanosecond of the simulation and is rather used to mean ‘per nanosecond’. Line 313: "6 hydrogen bonds with Arg176" - In which and how many of these hydrogen bonds does arginine act as an acceptor and in which as a donor? Which ligand atoms interact with Arg176? The same questions apply to Asn173. Ligand-protein interactions are stable due to the number of hydrogen bonds formed at a given time (up to 11) and how long these interactions last. Please check how long the given hydrogen bonds with Arg176 and Asn173 persist (its occupancy) during 100 ns of the simulation, i.e., for what percentage of the simulation time is it 'active'? Which of these H-bonds are the most stable?
Supplementary Information
In the section ‘Calculation of molecular mechanics of Poisson-Boltzmann surface area (mmpbsa)’ I propose adding an equation according to which the DG (deltaG) value is calculated and considering moving this section from the SI to the main text (3.5. Calculation of molecular mechanics of Poisson-Boltzmann surface area (mmpbsa)). Moreover, if ‘The large contribution of van der Waals interactions to the total binding energy suggests a hydrophobic contact of the ligand with the receptor’ it would be worthwhile to determine with which hydrophobic amino acids the substrate molecule interacts – see the remarks regarding “the hydrophobic nature of the pocket can be assumed” (point 3.3 of the manuscript).
Author Response
Comments and Suggestions for Authors
biotech-3197981-peer-review-v1
’Structural analysis and substrate specificity of D-carbamoylase from Pseudomonas’
Introduction
Comment 1[Lines 70-72: In which databanks are data on 3-dimensional crystal structures of wild-type and mutant D-carbamoylases deposited?]
Response 1 [Edited in the text (line 72).
Comment 2 [Lines 84-86: „Structural analysis of the enzyme's three-dimensional structure provides important information useful for production of the desired D-amino acid from D-hydantoins.” In the presented work, such a structural analysis was not performed. The structure of the modeled protein was not even shown, the location of the putative binding site was not indicated, and the amino acids that form the binding site were not listed.]
Response 2 [Edited in the text (lines 85-86).]
Comment 3 [2.7. Molecular docking
Please provide values for the parameters used during docking.]
Response 3 [Edited in the text (lines 192-194). The details are added in the chapter “2.7. Molecular docking”.]
Comment 4 [2.8. Molecular dynamics
Please provide information on the boundary conditions applied during the MD simulations.]
Response 4 [Edited in the text (lines 200-201). The details are added in the chapter “2.8. Molecular dynamics”.]
Comment 5 [3.3. Molecular docking of D-carbamoylase
The work is missing a figure with the three-dimensional structure of the modeled protein, including the putative binding site location.]
Response 5 [Edited in the text (lines 304-307). The Figure 3B is generated and added in the text of manuscript.]
Comment 6 [Near Figure 3 ... “the hydrophobic nature of the pocket can be assumed”. We do not have to assume that the binding site is hydrophobic. Discovery Studio Visualizer offers tools that can help determine the nature of the binding site, find hydrophobic amino acids, or show the surface of the binding site with mapped hydrophobicity.]
Response 6 [Edited in the text (lines 292-293). The picture 3B and text of the manuscript have been modified respectively.]
Comment 7 [Figure 3. Figure 3A should have a higher resolution. Figure 3B: The catalytically important amino acids Lys127 and Cys172, mentioned in line 270, should be shown in Figure 3B. Figure caption: The figure caption lacks an explanation of which color corresponds to which type of interaction – for example, we don’t know which are conventional hydrogen bonds and which are hydrophobic interactions. ‘Structure with dotted lines showing bonds and their lengths in angstroms’ suggests covalent bonds, whereas these are non-bonding intermolecular interactions, including but not limited to hydrogen bonds. Furthermore, covalent bond lengths and the distances over which intermolecular interactions occur are usually given to three decimal places, not five.]
Response 7 [Edited in the text (lines 305-307). Figure 3A has been replaced with a higher resolution figure. Lys127 and Cys172 are now visible in Figure 3B. The figure legend has been changed. The bond length distances have been modified.]
Comment 8 [Since the activity of this enzyme has been experimentally tested on several compounds (3.6. D-carbamoylase substrate specificity), including six ‘D-substrates’, I would like to see information (figures similar to Figure 3) on what poses these ligands adopt in the active site and in what interactions with the protein they participate. I mean here, for example, information obtained in docking, although MD simulations would be even more valuable.]
Response 8 [In this study, we selected only N-carbamoyl-D-tryptophan, a bulky amino acid, to evaluate/harmonize D-carbamoylase with our previously developed D-hydantoinase for further development of a two-enzyme system.]
Comment 9 [3.4. Molecular dynamics of D-carbamoylase binding to carbamoylated amino acids
The RMSD plot for the ligand (N-carbamoyl-D-tryptophan molecule) is missing – it should also stabilize in the active site.]
Response 9 [Edited in the text (lines 321-322). The RMSD figure was modified and RMSD of N-carbamoyl-D-tryptophan was added.]
Comment 10 [How do the authors explain the differences in RMSF for the complex and the APO protein? Does ligand binding stabilize or destabilize the structure of the D-carbamoylase? Where in the protein chain are the amino acids with higher RMSF? In the binding site, outside it, are they loops or helices? There is no information about amino acids' numbers (indices) constituting the active site. For the reader, the information that "for amino acids representing the binding cavity, the RMSF values were lower than or close to the mean RMSF value" is impossible to verify. It would also be interesting to compare the RMSF of the binding site in the APO form and the complex.]
Response 10 [Edited in the text (lines 323, 330-334). The manuscript text has been revised and a new Figure has been added showing the RMSF values ​​of the amino acids interacting with the ligand.]
Comment 11 [Lines 295-296: ‘To understand whether the simulated protein stably folds or unfolds over the course of a 100 ns simulation’ – Formally, we are not dealing with protein folding or unfolding. The changes in Rg and RMSD up to around 35-40 ns indicate the structural adaptation of protein to the new situation when there is a ligand in the active site. However, from about 70 ns, the RMSD and Rg graphs of the APO form and the complex become almost convergent. Therefore, if the binding site is sufficiently compact, one can also analyze the course of Rg changes during the simulation for the active site (amino acids that form it), not only the entire protein.]
Response 11 [Edited in the text (lines 336-339).]
Comment 12[ ‘1 ns’ in lines 313 and 314 suggests the first nanosecond of the simulation and is rather used to mean ‘per nanosecond’. ]
Response 12 [Edited in the text (lines 358, 360).]
Comment 13 [Line 313: "6 hydrogen bonds with Arg176" - In which and how many of these hydrogen bonds does arginine act as an acceptor and in which as a donor? Which ligand atoms interact with Arg176? The same questions apply to Asn173.]
Response 13 [Edited in the text (lines 360-364). The text of manuscript was modified respectively.]
Comment 14 [Ligand-protein interactions are stable due to the number of hydrogen bonds formed at a given time (up to 11) and how long these interactions last. Please check how long the given hydrogen bonds with Arg176 and Asn173 persist (its occupancy) during 100 ns of the simulation, i.e., for what percentage of the simulation time is it 'active'? Which of these H-bonds are the most stable?]
Response 14 [The Gromacs 2023.1 version we are currently using has technical issues in gmx hbond (memory), so unfortunately, we cannot provide additional information about hydrogen bonds.]
Comment 15 [Supplementary Information
In the section ‘Calculation of molecular mechanics of Poisson-Boltzmann surface area (mmpbsa)’ I propose adding an equation according to which the DG (deltaG) value is calculated and considering moving this section from the SI to the main text (3.5. Calculation of molecular mechanics of Poisson-Boltzmann surface area (mmpbsa)). Moreover, if ‘The large contribution of van der Waals interactions to the total binding energy suggests a hydrophobic contact of the ligand with the receptor’ it would be worthwhile to determine with which hydrophobic amino acids the substrate molecule interacts – see the remarks regarding “the hydrophobic nature of the pocket can be assumed” (point 3.3 of the manuscript).]
Response 15 [Edited in the manuscript text (lines 376-385). The information is moved to main part of the article and equation is added.]
Thank you very much for reviewing our manuscript.
Submission Date 23 August 2024
Date of this review 15 Sep 2024 11:06:09
Round 2
Reviewer 2 Report
Comments and Suggestions for Authors
All my concerns are addressed in revised version.
Author Response
Comment 1 [All my concerns are addressed in revised version.]
Response 1 [Thank you very much for reviewing our manuscript and for positive feedback.]
Reviewer 3 Report
Comments and Suggestions for Authors
Comments and Suggestions for Authors
biotech-3197981-peer-review-v2
’Structural analysis and substrate specificity of D-carbamoylase from Pseudomonas’
Concerns Response 6
Figure 3B has been modified, but the surface shown is dominated by blue – at least in this view, which suggests a predominance of negative hydrophobicity values, which would rather indicate a hydrophilic (polar) character of the binding site.
Concerns Response 7
Lys127 is visible, but Cys172 and its label are partially hidden. Please set the ‘Movable’ parameter in ‘Label Attributes’ to ‘Yes’ and slightly move the "Cys172" label to avoid overlapping with other labels and amino acids. In the ‘Interaction options’ of the Interaction monitor, I suggest turning off the display of intramolecular interactions – now we see hydrogen bonds between enzyme amino acids or the hydrophobic interaction His144-Pro199, which are irrelevant from the point of view of ligand-protein interactions. In the first version of the manuscript (biotech-3197981-peer-review-v1) Figure 3B correctly showed only intermolecular interactions.
Concerns Response 9
Unclear (complicated) Figure 4 caption.
Concerns Response 11
Lines 330-333 (real numbering in revised PDF file): adaptation, not adoption.
Concerns Response 15
Line 381-382: The large contribution of van der Waals interactions to the total binding energy suggests a hydrophobic contact of the ligand with the receptor (Figure 3B).
Although the MM/PBSA calculations indicate a large contribution of hydrophobic interactions to the binding free energy, the surface shown in Figure 3B is dominated by blue (suggesting low hydrophobicity), the text also does not list hydrophobic amino acids interacting with the ligand ('purple dotted lines' in Figure 3A/3B) or other hydrophobic amino acids in the vicinity of the bound ligand, not necessarily captured by the Discovery Studio interaction monitor.
Can we identify the amino acids involved in hydrophobic interactions with the ligand or the hydrophobic amino acids surrounding the ligand in the binding site and provide their names in the text? After all, based on Figure 3B in the first version of the manuscript, we see one hydrophobic interaction: the pi-alkyl interaction between the ligand and Pro199.
Author Response
Comment 1 [Concerns Response 6
Figure 3B has been modified, but the surface shown is dominated by blue – at least in this view, which suggests a predominance of negative hydrophobicity values, which would rather indicate a hydrophilic (polar) character of the binding site.]
Response 1 [Yes, the Figure 3B shows predominantly negative hydrophobicity, with positive incorporations. The text of the manuscript has been edited accordingly (lines 301-302).]
Comment 2 [Concerns Response 7
Lys127 is visible, but Cys172 and its label are partially hidden. Please set the ‘Movable’ parameter in ‘Label Attributes’ to ‘Yes’ and slightly move the "Cys172" label to avoid overlapping with other labels and amino acids. In the ‘Interaction options’ of the Interaction monitor, I suggest turning off the display of intramolecular interactions – now we see hydrogen bonds between enzyme amino acids or the hydrophobic interaction His144-Pro199, which are irrelevant from the point of view of ligand-protein interactions. In the first version of the manuscript (biotech-3197981-peer-review-v1) Figure 3B correctly showed only intermolecular interactions.]
Response 2 [The Figure 3B is edited: Cys172 is now visible and the intramolecular interactions are hidden.]
Comment 3 [Concerns Response 9
Unclear (complicated) Figure 4 caption.]
Response 3 [Edited in the text (lines 317-321)]
Comment 4 [Concerns Response 11
Lines 330-333 (real numbering in revised PDF file): adaptation, not adoption.]
Response 4 [Edited in the text (lines 337, 339)]
Comment 5 [Concerns Response 15
Line 381-382: The large contribution of van der Waals interactions to the total binding energy suggests a hydrophobic contact of the ligand with the receptor (Figure 3B).
Although the MM/PBSA calculations indicate a large contribution of hydrophobic interactions to the binding free energy, the surface shown in Figure 3B is dominated by blue (suggesting low hydrophobicity), the text also does not list hydrophobic amino acids interacting with the ligand ('purple dotted lines' in Figure 3A/3B) or other hydrophobic amino acids in the vicinity of the bound ligand, not necessarily captured by the Discovery Studio interaction monitor.
Can we identify the amino acids involved in hydrophobic interactions with the ligand or the hydrophobic amino acids surrounding the ligand in the binding site and provide their names in the text? After all, based on Figure 3B in the first version of the manuscript, we see one hydrophobic interaction: the pi-alkyl interaction between the ligand and Pro199.]
Response 5 [Edited in the text (lines 388-392)]
Thank you very much for reviewing our manuscript and suggesting ways to overcome the questions that arose.
Please be informed that Prof. Sakanyan and Dr. Koloyan were unable to upload the current response. Therefore, I am sending you our response, as I am also a corresponding author.
Hovsep Aganyants